# Antiviral Effect of Antimicrobial Peptoid TM9 and Murine Model of Respiratory Coronavirus Infection

**DOI:** 10.3390/pharmaceutics16040464

**Published:** 2024-03-27

**Authors:** Maxim Lebedev, Aaron B. Benjamin, Sathish Kumar, Natalia Molchanova, Jennifer S. Lin, Kent J. Koster, Julian L. Leibowitz, Annelise E. Barron, Jeffrey D. Cirillo

**Affiliations:** 1School of Medicine, Texas A&M University, Bryan, TX 77807, USA; maxim.i.lebedev@gmail.com (M.L.); aaronbbenjamin@tamu.edu (A.B.B.); viralkskumar@gmail.com (S.K.); kkoster@tamu.edu (K.J.K.); jleibowitz@tamu.edu (J.L.L.); 2Department of Bioengineering, Stanford University, Stanford, CA 94305, USA; nmolchanova@lbl.gov (N.M.); jlin3@stanford.edu (J.S.L.); aebarron@stanford.edu (A.E.B.); 3Molecular Foundry, Lawrence Berkeley National Laboratory, Berkeley, CA 94720, USA

**Keywords:** peptoid, MHV, SARS-CoV-2, antimicrobials

## Abstract

New antiviral agents are essential to improving treatment and control of SARS-CoV-2 infections that can lead to the disease COVID-19. Antimicrobial peptoids are sequence-specific oligo-*N*-substituted glycine peptidomimetics that emulate the structure and function of natural antimicrobial peptides but are resistant to proteases. We demonstrate antiviral activity of a new peptoid (TM9) against the coronavirus, murine hepatitis virus (MHV), as a closely related model for the structure and antiviral susceptibility profile of SARS-CoV-2. This peptoid mimics the human cathelicidin LL-37, which has also been shown to have antimicrobial and antiviral activity. In this study, TM9 was effective against three murine coronavirus strains, demonstrating that the therapeutic window is large enough to allow the use of TM9 for treatment. All three isolates of MHV generated infection in mice after 15 min of exposure by aerosol using the Madison aerosol chamber, and all three viral strains could be isolated from the lungs throughout the 5-day observation period post-infection, with the peak titers on day 2. MHV-A59 and MHV-A59-GFP were also isolated from the liver, heart, spleen, olfactory bulbs, and brain. These data demonstrate that MHV serves as a valuable natural murine model of coronavirus pathogenesis in multiple organs, including the brain.

## 1. Introduction

SARS-CoV-2 causes COVID-19, a disease characterized by a fever, a cough, shortness of breath, and pneumonia. The disease can be fatal in vulnerable individuals, including immunocompromised individuals and the elderly. SARS-CoV-2 can be transmitted via aerosol and through contact with contaminated surfaces [1]. The search for and validation of new potential antiviral chemical and biological agents is essential for the improvement in treatments and control of severe acute respiratory syndrome (SARS). This includes new variants, which arise due to mutations, which render other therapeutics and vaccines less effective than before.

Currently, there are many directions being tested towards the development and application of therapeutics against SARS-CoV-2. These include antiviral compounds targeting virion structures such as spike protein [2], the inhibition of RNA-dependent RNA polymerase by remdesivir [3] and favipiravir [4], the inhibition of proteases like SARS-coronavirus papain-like protease (PLpro) and 3C-like protease (3Clpro) [5,6], and targeting of type I and type III interferons [7]. The use of virus-neutralizing antibodies is another effective approach against SARS-CoV-2. Polyclonal antibodies in the form of convalescent serum as well as monoclonal antibodies have been shown to be effective towards controlling infections, including SARS-CoV-2 [8,9]. Another interesting direction is the utilization of clustered regularly interspaced short palindromic repeats (CRISPR) technology as a therapeutic to target the viral genome and destruction of specific viral RNA with Cas13. This approach, called PACMAN (prophylactic antiviral CRISPR in human cells), has been developed, albeit with many limitations such as delivery [10].

Most of these approaches are still in various stages of development and have either not been approved or have emergency use authorization only. Others have more prominent side effects, low availability, and high costs in addition to other limitations. In addition to these issues, there has been significant issues towards the creation of novel classes of antimicrobial drugs in the past 30 years. While some advancements have been seen, challenges specifically with antivirals are apparent. Of the more than 220 viruses that infect humans, only 10 have antiviral drugs that are clinically approved [11]. Therefore, further drug discovery efforts are needed to address this problem. Here, we propose the antimicrobial peptide mimic, peptoids, as another approach to curtail an infection. There are many types of mimics used as antimicrobials, leading to a wide variety of different mechanisms and ways to target viruses and bacteria [12,13]. These mimics are often inspired by nature, including mimics of peptides like the magainins or Histatin 5 [12,13]. Most of these mimics use substitutions of specific amino acids to enhance the therapeutic effectiveness of the naturally occurring antimicrobial peptide. When mimicking antimicrobials found in nature, certain factors will be kept similar or as close to the natural molecule. These factors include shape, size, charge, and others. As such, it is possible to take advantage of properties to target specific areas of the bacteria like the membrane, such as the fact that α-helices tend to undergo hydrophobic interactions, or lipopeptides using lipids [14,15,16]. Some examples of antimicrobials attempt to take advantage of multiple factors or try to shorten naturally occurring antimicrobial peptides (AMPs). One such example is innate defense regulator peptides, which are AMPs that have been optimized for immunomodulatory function [17]. While originally used for a different function, it was also discovered that one such mimic, IDR-1, had activity against MRSA and VRSA [17]. Another mimic-based approach is the use of peptoids. Peptoids are sequence-specific poly-*N*-substituted glycines, mimicking the function of AMPs with the added benefit of being resistant to proteases [18]. It is well known that AMPs such as defensins and cathelicidins are a part of the innate defense against infections [19], including the sole human cathelicidin, LL-37 [20,21]. To the best of our knowledge, the mechanism of the antimicrobial action of AMPs is based on the disruption of the cell membrane, either by pore formation or the alteration of lipid layer structure and function [22,23], or through intracellular effects such as inhibiting transcription, translation, or protein synthesis [24,25]. Peptoids, as non-natural forms of AMPs or mimetics of AMPs, have demonstrated similar antimicrobial properties. One direct advantage of antimicrobial peptoids over antimicrobial peptides is the ability to resist protease degradation due to the movement of the R group from the α-carbon to the amide [26]. Additionally, peptoids have been shown to act quickly in killing bacteria in vitro, especially compared to traditionally used antibiotics [27].

Several different peptoids have been shown to be effective against various Gram-positive and Gram-negative bacteria [28,29] and viruses, including herpes simplex virus type 1 and SARS-CoV-2 [30,31]. Among the peptoids that have been tested against these two viruses, TM9, or MXB-9, demonstrated the best antiviral effect [30]. The predicted mechanism of action for the TM9 peptoid against enveloped viruses is the disruption of the envelope membrane structure, although this has not yet been fully elucidated [31]. Murine hepatitis virus and SARS-CoV-2 are both members of the genus Betacoronavirus [32] and have similar genome structure and biology. Humans are not susceptible to MHV infection, making it safer and easier to perform testing with MHV since it does not require a BSL-3 laboratory setting [33,34,35]. Virulence and organotropism of MHV depends on the virus strain and varies as multiple strains exist. Depending upon the tropism of the particular strain used, this virus serves as a model for human respiratory, gastrointestinal, liver, and neurologic infections [33].

In this study, we use three coronavirus strains, MHV-1, MHV-A59, and recombinant MHV-A59-GFP, to examine respiratory system tropism and the impacts of virulence on therapeutic efficacy. The MHV coronaviruses have been shown to generate severe acute respiratory syndrome (SARS) in mice that demonstrates pathological similarities to human SARS [36,37]. We observed antiviral activity of a novel peptoid TM9 for all three MHV strains, suggesting that TM9 will be effective against all coronaviruses, including SARS-CoV-2. We utilize an aerosol route of infection for MHV in mice via a Madison chamber that produces a dry aerosol, making this model closely mimic the natural route of SARS-CoV-2 transmission [38,39] and confirming that MHV represents an inexpensive, rapid, and straightforward model for COVID-19 therapeutic screening.

## 2. Materials and Methods

### 2.1. Cell Culture

Murine lung-carcinoma-derived cell line L2 lung-carcinoma-derived fibroblastic cells [40], human A549 (human lung carcinoma (ATCC CCL-185, ATCC, Manassas, VA, USA)), and HULEC-5a (human microvascular endothelium (ATCC CRL-3244)) cell lines were used in this study. Cells were cultured in cell culture dishes with Dulbecco’s Modification of Eagle’s Medium (DMEM) (Corning, Corning, NY, USA) containing 4.5 g/L of glucose, L-glutamine, and sodium pyruvate and supplemented with 10% of FBS, 100 U/mL of penicillin, 10 μg/mL of streptomycin (Lonza, Basel, Switzerland), GlutaMax (Gibco, Waltham, MA, USA), a 0.01 M HEPES buffer solution (Gibco), and MEM nonessential amino acids (Corning).

### 2.2. Synthesis of TM9 Peptoid

The peptoid was synthesized as described previously [27,30,41]. Briefly, peptoid synthesis was carried out using a Symphony X (Gyros Protein Technologies, Tucson, AZ, USA) peptide synthesizer located at the Molecular Foundry in the Lawrence Berkeley National Laboratory, Berkeley, CA, USA. Peptoids were synthesized on a Rink amide MBHA resin (EMD Biosciences, Gibbstown, NJ, USA). All reagents were purchased from Sigma Aldrich (St. Louis, MO, USA). Synthesis followed the submonomer protocol from Zuckermann, et al. [41]. Peptoids were cleaved from the resin by treating with trifluoroacetic acid (TFA)/triisopropylsilane/water (95:2.5:2.5 volume ratio) for 30 min. A C18 column in a reversed-phase high-performance liquid chromatography (HPLC) system (Waters Corporation, Milford, MA, USA) was used for purification with a linear acetonitrile and water gradient with a compound purity greater than 98% as measured by analytical reverse-phased HPLC. The structure of TM9 is seen in Appendix A. The confirmation of the peptoid synthesis was determined using electrospray ionization mass spectrometry (predicted—1271.12 g/mol, measured—1271.50 g/mol) (Appendix A).

### 2.3. Test of Efficacy and Toxicity of TM9 Peptoid

In total, 15,000 cells/well of L2, A549, and HULEC-5a cell lines was plated in 96-well plates with 5 replicates for each concentration of peptoid and incubated at 37 °C, 5% CO_2_, for 24 h. After incubation, 10 two-fold serial dilutions of TM9 were prepared from 0.98 μg/mL to 500 µg/mL. Each dilution of TM9 was added to respective wells with cell cultures and to separate plates with viruses. The cells and viruses were incubated for 2 h. Then, preincubated viruses with MOI: 0.1 were added to the cells containing respective concentrations of peptoids and incubated for 3 h and 40 h. After incubation, the supernatant of each replicate was collected for a plaque assay.

Cell viability was measured using a 10% alamarBlue-containing growth medium. It was added to each well and incubated for 2 h. After the incubation, the alamarBlue-containing medium was transferred to a new clear 96-well plate and the absorbance was measured at 490 nm. Viability was also measured, as mentioned previously, using cells that had been exposed to the virus at an MOI of 0.1 for 2 h. These two measurements were compared to each other to yield the change in %viability.

### 2.4. Virus

The following three strains of the murine hepatitis virus were used: MHV-1 (ATCC VR-261) and MHV-A59 (ATCC VR-764) and MHV-A59-GFP. MHV-A-59-GFP is a recombinant version of MHV-A59 generated by Susan Weiss’s lab where the ns4 gene has been replaced by the EGFP sequence [42]. Stock viruses were stored at −80 °C.

### 2.5. Plaque Assay

L2 cells were plated in 12-well plates at 0.4 × 10^6^ cells per well. Cells were seeded in complete DMEM with 10% FBS, 1 mL per well, and incubated at 37 °C for two days. Serial ten-fold dilutions of the virus in DMEM in 5 mL snap cap tubes were prepared. DMEM was aspirated from the L2 cell monolayers and 200 µL of virus dilutions was added to each well. Plates were incubated with the virus inoculum at room temperature for 60 min with periodic shaking. After incubation, 1.6% agarose melted and cooled to 50 °C was mixed with the same volume of 2× EMEM (Quality Biological, Gaithersburg, MD, USA) to make an overlay solution, 1 mL of which was added to each well. After adding overlays, cells were incubated for 2 days at 37 °C, 5% CO_2_ and humidity. After the incubation, overlays were carefully removed and cells were fixed with 10% neutral buffered formalin (Sigma) for 15 min, stained with 0.1% crystal violet in 20% ethanol for 5 min, and washed.

### 2.6. Generation of the Animal Model for Respiratory Infection

5–7-week-old female BALB/c (Envigo, Indianapolis, IN, USA) mice were used as a model of the infection according to the protocol approved by Texas A&M Animal Care and Use Committee. Animals were randomly distributed into 4 groups: group 1—5 animals as the uninfected control group; group 2—10 mice infected with MHV-1; group 3—10 mice infected with MHV-A59; group 4—10 mice infected with MHV-A59-GFP. The Madison aerosol chamber was used to infect the animals. Viral supernatants in DMEM were used in the following titers: MHV-1—10^5^ PFU/mL; MHV-A59—10^7^ PFU/mL; MHV-A59-GFP—10^7^ PFU/mL. Each group of mice was exposed to the corresponding virus for a total of 15 min by running 3 consecutive 5 min infection cycles. Clinical observations were made before infection and every 12 h for 5 days after the infection.

Every 24 h, two animals from each experimental group and one animal from the control group were sacrificed. Necropsy and gross pathological examination were performed. Lungs, the brain, olfactory bulbs, the heart, the liver, the spleen, and kidneys were homogenized with 1 mL of DMEM. Homogenates were used for the plaque assay to quantify the virus.

### 2.7. Statistical Analysis and Graphics

Statistics and data management were developed using Microsoft Excel Worksheets and Graphpad Prism 10.2.2 software.

## 3. Results

### 3.1. The Effect of TM9 Peptoid Was Similar on Different Cell Lines but Dependent on the Duration of Exposure

Both a murine lung-carcinoma-derived cell line (L2) and two types of human cells, lung carcinoma (A549) and microvascular endothelial cells (HULEC-5a), demonstrated similar sensitivity to the peptoid TM9 3 h post-incubation (Figure 1). The cytotoxic concentration with 50% viability (CC50) was measured to be between 20 and 30 μg/mL for all cell lines. The cytotoxic concentration where there was 90% cell viability (CC90) was measured at 81.34 μg/mL for L2 cells, 60.97 for A549 cells, and 36.43 μg/mL for HULEC-5a cells.

Comparing the 40 h time point (Figure 2) to the 3 h time point, the CC50 reduced by 43% in L2 cells, down to 17.83 μg/mL, and the CC90 reduced to 31.54 μg/mL. For A549 cells, the CC50 decreased to 16.68 μg/mL while the CC90 reduced to 25.42 μg/mL. Lastly, HULEC-5a cells showed a reduction in the CC50 to 10.31 μg/mL and CC90 to 14.23 μg/mL. The ratio of CC90 to CC50 was almost three for both the murine lung carcinoma cells and human lung carcinoma cells after three hours. Human intravascular endothelial cells were more sensitive to the peptoid as these cells had the lowest ratio (1.81) of CC90 to CC50. For all three cell lines, exposure to TM9 for 40 h resulted in greater cytotoxic effects at lower concentrations of the peptoid and caused the ratio of CC90 to CC50 to decrease.

### 3.2. The Effective Antiviral Concentration of TM9 Was within a Therapeutic Window in L2 Cell Culture

L2 cells were preincubated with serial dilutions of TM9 before infection with either MHV-1, MHV-A59, or MHV-A59-GFP (Figure 3). The CC50 of L2 cells infected with MHV-1 was 17.52 μg/mL, while the effective concentration of the peptoid against the virus (EC50) was 4.47 μg/mL, yielding a therapeutic index (CC50/EC90) of 3.92. MHV-A59-infected cells yielded a CC50 of 29.30 μg/mL and an EC-90 of 15.92 μg/mL, resulting in a therapeutic index of 1.84. Lastly, L2 cells infected with MHV-A59-GFP had a CC50 of 29.28 μg/mL and an EC90 of 8.99 μg/mL, resulting in a therapeutic index of 3.25, the highest among all three viruses. The effect of individual viruses on the cytotoxicity of TM9 is seen in Figure 3D. MHV-1 saw a decrease in CC50 and CC90, while both MHV-A59 and MHV-A59-GFP saw large increases in the CC50.

### 3.3. Respiratory MHV Infection Was Effectively Generated in All Experimental Animals by Aerosol Inoculation Using Madison Chamber

Mice were infected with one of the three MHV viral strains using a Madison chamber at 10^5^ PFU/mL for MHV-1 and 10^7^ PFU/mL for both MHV-A59 and MHV-A59-GFP. Mice that were exposed to aerosols of one of these viruses demonstrated signs of the disease starting from the first day post-infection, lasting up to the third day post-infection. Clinical signs of the infection, including lethargy, dyspnea, nasal discharge, and a fever (39–40 °C), were observed throughout these three days, subsiding afterwards.

To determine the effect that the virus had on these mice, two mice from each group were euthanized daily. Subsequently, necropsies were performed to determine the pathological effect on the mice and the viral titers seen in various organs. Necropsy revealed pathological changes in the lungs including swelling in lung tissue and redness due to vascular dilation. There were no visible gross-pathological changes in other organs during the entire observation period.

Viral loads were 100-fold and 10-fold lower in lung homogenates for MHV-1 than those observed for both MHV-A549 strains starting from day one post-infection (Figure 4). All mice treated with the virus increased viral loads by day two post-infection, with increases approximately one log higher than on day one. Viral titers decreased from this point on until the final day of observation, day 5, resulting in a final decrease of one log compared to the original titers of day 1. All MHV-1 viral loads were two logs lower than both MHV-A59 strains. Interestingly, another difference seen between these strains was the ability for systemic dissemination to other organs. In fact, the only other organ that showed the MHV-1 virus as detected and quantified by the plaque assay was the olfactory bulbs, seen exclusively on day 5 post-infection. Conversely, the MHV-A59 virus was detected in olfactory bulbs, the liver, the heart, and the spleen on day two post-infection. From day 3 onward, this strain was detectable in the brain, with a significant increase in viral loads (2 logs) on day 4. MHV-A59-GFP was initially detected in olfactory bulbs on day 3 post-infection. From here onward, the viral load increased until the end of the observation period. The virus was detectable in the brain and heart starting on day 4 and in the liver starting on day 5.

## 4. Discussion

In this work, we determined the therapeutic window and therapeutic index for the TM9 peptoid in vitro. TM9 has been shown to have a similar mechanism to the human cathelicidin LL-37. It has been demonstrated that LL-37 causes disruption in the cell membrane of bacteria, causing pore formation, which leads to cytoplasmic leaking, osmotic disbalance, and multiple functional dysregulations [43]. As such, we can speculate that the effect of this peptoid on the membrane of the viral envelope is similar to its effect on the bacterial membrane (Figure 5). While this model needs further investigation, we believe that the similarities of virion envelope structure with bacterial membrane structure likely mean that TM9 acts against virions in a similar manner to how it acts against bacteria. There are several potential ways in which TM9 may affect the ability of MHV virions to be viable. The first is that TM9 prevents virions from being able to attach to the host cell receptors by blocking or binding to receptors on the virion. If this were to be the case, virions would be unable to bind to the surface receptors of the cells they were infecting, resulting in reduced ability to infect cells. A second possibility is that TM9 binds to the membrane component of the virion and exposes the internal parts of the virion to an extracellular environment. This would result in lowered viability as virions’ sensitive RNA genome would be exposed to any RNAses in the media or have key components leaking into the environment, likely before they are able to infect cells. As the virion envelope is typically derived from the mammalian cell outer or inner membrane and has similar structure, this model likely is involved. It is possible that both mechanisms are occurring simultaneously. If this is the case, the most important question is how the antiviral effect of the peptoid is related to its cytotoxic effect. Recently, it has been shown that phosphatidylserine in the viral envelope plays an important role in membrane susceptibility to the disruption by the peptoid, and may thus be a potential way to optimize future peptoid design towards virions over mammalian cells [31].

Here, we demonstrated that the effect of the TM9 peptoid on mammalian cells showed a difference between cytotoxic concentration and maximum nontoxic concentration. Both murine and human lung carcinoma cells had similar levels of tolerance to TM9. According to our results, the CC_50_ of TM9 3 h post-incubation was 20–30 μg/mL for both lung carcinoma (A549) cells and L2 cells, while the CC_90_ was approximately 61 μg/mL and 81 μg/mL, respectively, while microvascular endothelial cells had a CC_90_ of approximately 36 μg/mL. This suggests that human intravascular endothelial cells were more sensitive to the peptoid, which should be considered upon the parenteral administration of the peptoid. Other studies demonstrated that tumor cells can be more sensitive to cationic antimicrobial peptides due to higher metabolic activity of these cells [44].

Cultured cells have the distinct disadvantage of being less accurate measures of cytotoxicity as they are missing many of the tight junctions involved in tissues. As such, as the next step in this work, it may be necessary to try alternative methods of a cell culture to obtain a better understanding of how cytotoxic TM9 is. One such method would be to perform testing via the air–liquid interface (ALI), which has been shown to cause changes to the surface properties of alveolar epithelial cells [45]. Subsequent or concurrent experiments in vivo may show even less toxicity and will provide insight into potential adverse systemic effects. Local application in the form of aerosol inhalation should also be considered as an alternative or additional therapeutic administration procedure. Local aerosol administration has demonstrated promising results in the application of antimicrobial peptides against other infections of the respiratory system [46,47].

The therapeutic index of TM9, as a ratio of CC50 to EC50, depends on the virulence of the viral strain according to our results. MHV-A59 was the most virulent strain and the effective concentration of TM9 against this virus was the highest. MHV-1 was the most susceptible to the peptoid and demonstrated the highest therapeutic index. Further investigation is required as TM9 demonstrated a relatively low therapeutic index in the traditional cell culture, particularly in the case of the most virulent strain of MHV. Therefore, cytotoxicity tests using only a cell culture model are likely not sufficient. In this case, a more comprehensive and integrative approach is required, including hemolytic activity assays, cytotoxicity tests in vivo, and apoptosis tests. This will all have to be performed using peripheral blood mononuclear cells as it was proposed previously to determine the therapeutic index for antimicrobial peptides [48] or in vitro using a 3D cell culture [28,30].

The second part of this work was to evaluate MHV infection as a model of human SARS-CoV-2 infection for further tests of peptoids and other antivirals in vivo. An important part of this goal was to determine if the utilization of the Madison chamber was an effective method to test aerosolized MHV in mice. Madison aerosol chambers produce a dry aerosol, and the animals present are not under anesthesia, allowing for the most natural transmission of the virus in a highly accurate and reproducible manner, providing a model of respiratory coronavirus infection and testing of antimicrobial peptoids and other antiviral therapeutics. Our results have shown that the utilization of the Madison chamber is an effective instrument for the aerosol inoculation of mice with respiratory MHV variants. All animals showed signs of infection via respiratory and/or other symptoms as is typically seen with MHV infection. Disease severity was dependent on the MHV strain; however, disease manifestations were observed in all infected mice, with the highest levels in animals infected with MHV-A59. Infection caused by MHV-A59-GFP was similar to infections with the MHV-A59 wild type based on viral load titers and their dynamics of change over time, but showed slower dissemination to other organs. Previous reports have shown that MHV-A59 tropism to multiple organs occurs, including infection of the respiratory tract [37] and the brain/liver [49]. MHV-1 infection was not systemic and was only detected in the respiratory tract and the olfactory bulb. This strain demonstrated lower virulence and is considered a strain with only respiratory tropism. As a result, it is often used in murine respiratory coronavirus models, but only in highly susceptible mice lineages such as A/J mice [50,51]. All of this suggests that the use of a Madison chamber for infection with MHV strains is not only possible, but effective. As MHV and SARS-CoV-2 are both Betacoronaviruses, it stands to reason that future studies with SARS-CoV-2 may be able to take advantage of the effectiveness of using a Madison chamber as well.

These data demonstrate that the effective concentration of TM9 is within the therapeutic range and that MHV provides a rapid, simple, and effective model for coronavirus infection and the evaluation of therapeutics in a systemic and organ-specific manner. Since these data are quite promising, further studies are warranted to evaluate the therapeutic efficacy of TM9 and other, more tailored peptoids in vivo and to use other coronaviruses, including SARS-CoV-2.

## Figures and Tables

**Figure 1 pharmaceutics-16-00464-f001:**
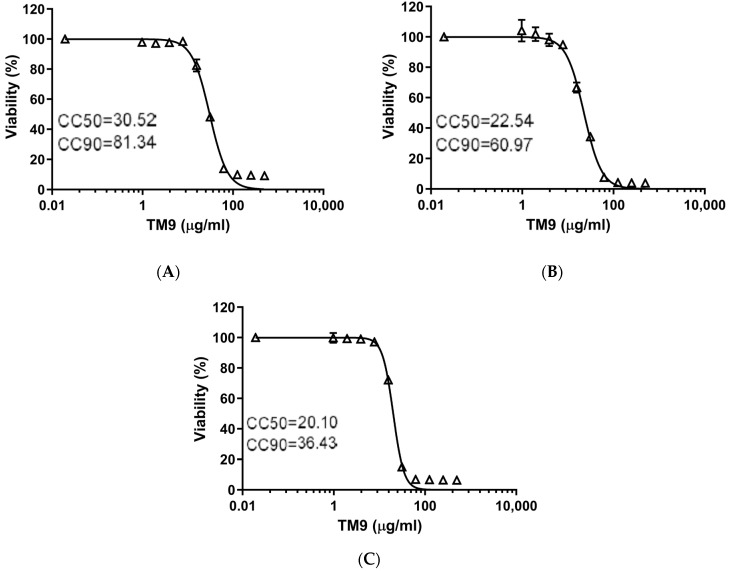
Toxicity test of TM9 peptoid in vitro. Each cell type was incubated for 3 h in cell culture medium with 10 two-fold serial dilutions of TM9. After the incubation, cell viability was measured using 10% alamarBlue reagent in growth medium (2 h). Absorbance was measured at 490 nm. Cell viability was calculated as a percentage of metabolic activity of corresponding untreated controls. Error bars show standard deviation of 5 experimental replicates. (**A**) L2 cells; (**B**) A549 cells; (**C**) HULEC-5a cells. CC50—cytotoxic concentration reducing viability by 50%; CC90—cytotoxic concentration reducing viability by 90%.

**Figure 2 pharmaceutics-16-00464-f002:**
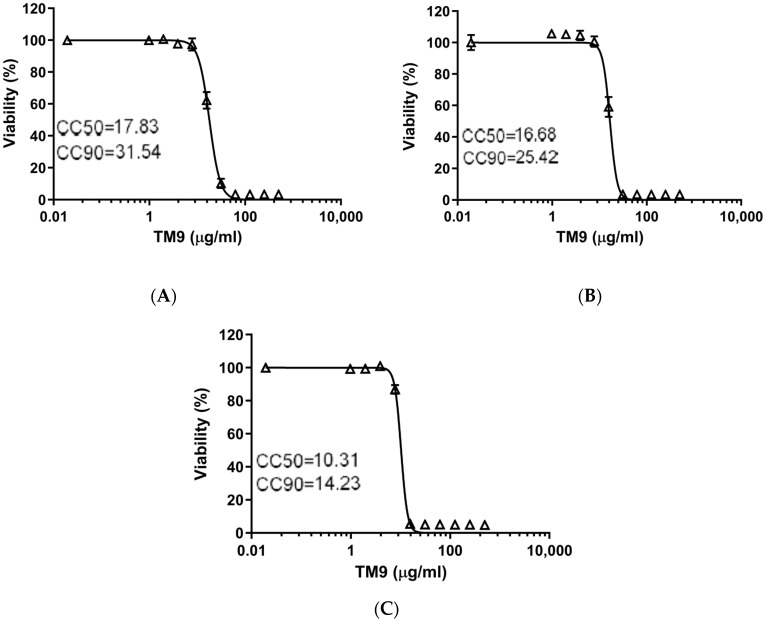
Prolonged toxicity test of TM9 peptoid in vitro. Cells were incubated with twofold serial dilutions of the peptoid in cell culture medium for 40 h. Cell viability was measured using 10% alamarBlue reagent. Cell viability was calculated as a percentage of metabolic activity of corresponding untreated controls. Error bars show standard deviation of 5 experimental replicates. (**A**) L2 cells; (**B**) A549 cells; (**C**) HULEC-5a cells. CC50—cytotoxic concentration reducing viability by 50%; CC90—cytotoxic concentration reducing viability by 90%.

**Figure 3 pharmaceutics-16-00464-f003:**
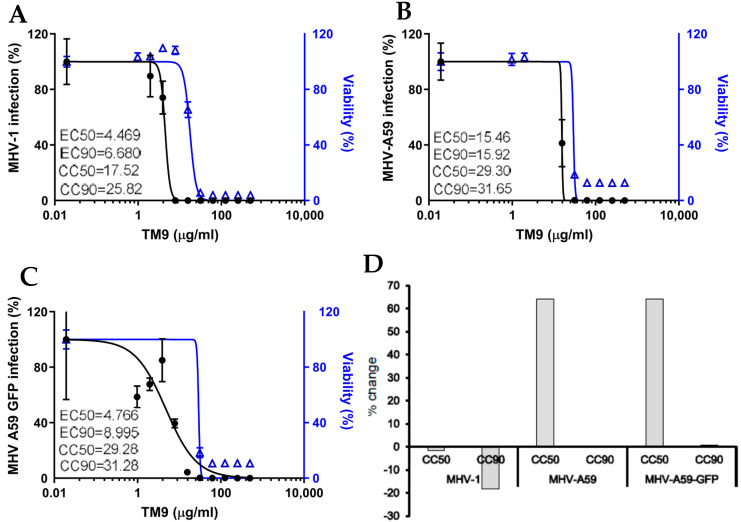
Therapeutic window in vitro. L2 cells and viruses were preincubated separately with TM9 peptoid for 3 h before infection. TM9-treated cells were infected with TM9-treated virus. After 40 h of post-infection incubation, cell culture medium was harvested and tested for the virus titer using plaque assay. Cells were incubated for 2 h with 10% alamarBlue in cell culture medium and absorbance was measured at 490 nm. Cell viability was calculated as a percentage of metabolic activity of corresponding untreated controls. Error bars show standard deviation of 5 experimental replicates. (**A**) MHV-1; (**B**) MHV-A59; (**C**) MHV-A59-GFP; (**D**) effect of the virus infection on cytotoxic concentration of TM9, or percentage of change in cytotoxic concentration in the presence of virus. EC50—effective concentration inactivating 50% of the virus; EC90—effective concentration inactivating 90% of the virus; CC50—cytotoxic concentration reducing viability by 50%; CC90—cytotoxic concentration reducing viability by 90%.

**Figure 4 pharmaceutics-16-00464-f004:**
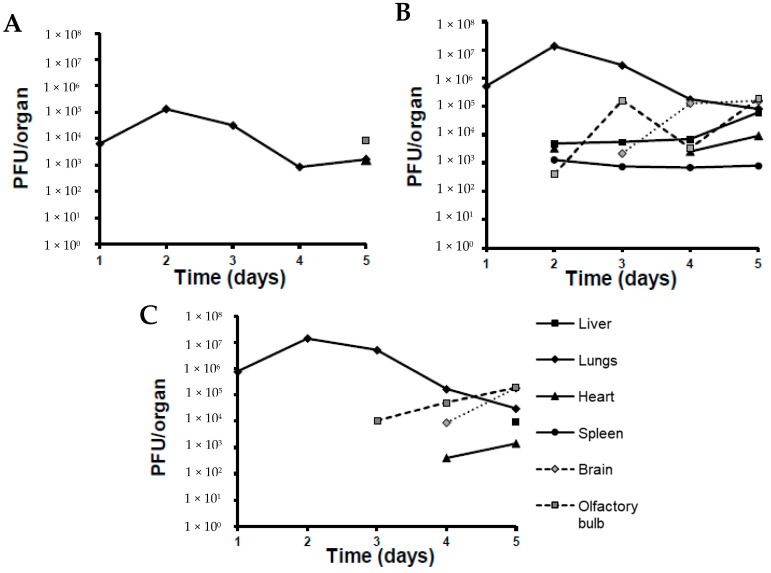
Viral loads in organs for 5 days post-infection. Corresponding groups of mice were infected with each virus using Madison aerosol chamber. Two animals from each group were sacrificed daily and viral titers were measured in their organ homogenates using plaque assay for 5 days post-infection (duration of the experiment). Each data point represents average number of plaque-forming units per organ from 2 animals. (**A**) MHV-1; (**B**) MHV-A59; (**C**) MHV-A59-GFP.

**Figure 5 pharmaceutics-16-00464-f005:**
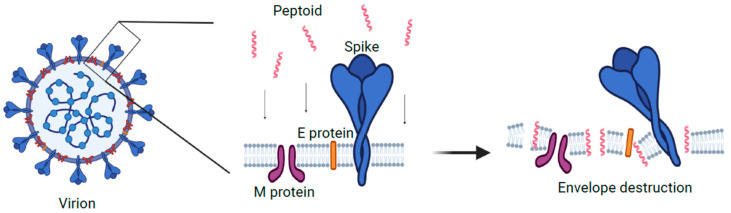
One of the proposed mechanisms of action for TM9 peptoid. Peptoid molecules destabilize structure and organization of the lipid bilayer and destroy viral envelope and/or disrupt its functions. Created with BioRender.com.

## Data Availability

The data presented in this study are available in this article (and Appendix A).

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
