# Peer review of "Antiviral Effect of Antimicrobial Peptoid TM9 and Murine Model of Respiratory Coronavirus Infection"

_pharmaceutics, 2024, doi:10.3390/pharmaceutics16040464_

Round 1

Reviewer 1 Report

Comments and Suggestions for Authors

See attached file.

Comments on the Quality of English Language

Author Response

Major:

  1. The title of the work suggests “peptoids”. However, only one peptoid, TM9 is used throughout the work. May be, it is better to say “of an antimicrobial peptoid”?

Replaced “peptoids” with “peptoid TM9”

  1. Confirmation of the peptoid synthesis (lines 118-119). Please, add the data on the predicted and measured mass of the peptoid.

Added the data showing that the peptoid is confirmed in the supplemental (Figure S2).

  1. Concerning Figure 5. Two mechanisms are discussed (line 307): 1) TM9 prevents virions from being able to attach to the host cell receptors; 2) a second possibility is that TM9 binds to the membrane component of the virion. In fact, only the second mechanism is illustrated in Fig. 5. Please, modify the legend of the figure, mentioning that this one of the proposed mechanisms. Otherwise, please, add illustration of the first variant.

Switched to “One of the proposed…”

  1. Also, concerning legend of Figure 5, please eliminate the word model. Et is enough to say; “Proposed mechanism…”.

Removed the word model.

Minor:

  1. Line 26-27: no keywords listed.

Added Keywords

  1. Line 78, reference 24, no mentioning TM9 peptoid, please specify the reference.

MXB-9 is the same peptoid, just called by a different name. Modified the text so that it is clear.

  1. This sentence is difficult to understand (lines 294-296): “While this model needs further investigation, we believe that the similarities of virion envelope structure likely means that TM9 acts against virions in a similar manner to how it acts against bacteria.” Which similarity you mean, between bacterial plasma membrane and virion envelope? Please, say more definitely.

Clarified by stating that we are talking about the similarities between both membranes of bacteria and virion envelope.

  1. Not clear to me, why some references (17, 23, 45) possess information on citation. These are internet resources. Should be “accessed on…” instead of “cited”?

Replaced cited with accessed on.

Reviewer 2 Report

Comments and Suggestions for Authors

Overall the manuscript is promising, but it feels a bit preliminary.   Specifically, the authors do not present any data that indicates viral membrane disruption is the mechanism of action.  While the authors clearly state that the mechanism is speculative, membrane permeabilization can be addressed through numerous experimental approaches which would give significant strength to the article.   Additionally, there is no mention of peptoid-control experiments, either with traditional a-peptides as a comparator, or other inactive peptoids to show this is a specific mechanism of TM9.

Other minor suggestions are below:

1. The figures in general are hard to see/read and would benefit from improved resolution or remaking them.  Specifically figures 1&2 are very small and would benefit from enlargement.  The authors can consider graphing all 3 cell types on one axis, with the appropriate parameters listed in a separate table or on the sides.  The panel designations (A,B,C) are entirely too small.

2. the introductory paragraph on CRISPR should be merged with the previous paragraph about alternative approaches.

3. Peptoids are only one of numerous types of AMP-mimetic compounds which act through membrane disruption and show therapeutic potential.  These should be at least mentioned as alternatives in the introduction with appropriate references.

4.  The figure legends are a bit confusing regarding the cell types.  Either the types of cells used should be more prominent in the legends (i.e. capitalized and parenthetical references to the panels) and/or included in the figures themselves.  

5. The authors should include a structure of TM9.

Author Response

    Overall the manuscript is promising, but it feels a bit preliminary.   Specifically, the authors do not present any data that indicates viral membrane disruption is the mechanism of action.  While the authors clearly state that the mechanism is speculative, membrane permeabilization can be addressed through numerous experimental approaches which would give significant strength to the article.  Additionally, there is no mention of peptoid-control experiments, either with traditional a-peptides as a comparator, or other inactive peptoids to show this is a specific mechanism of TM9.

We agree that there are experiments which need to be performed to elucidate the exact mechanism of action, however, those experiments are outside of the scope of this manuscript. We plan to do those experiments in a subsequent follow-up paper which will discuss more of the mechanism as opposed to the foundational work presented in this manuscript.

Other minor suggestions are below:

  1. The figures in general are hard to see/read and would benefit from improved resolution or remaking them.  Specifically figures 1&2 are very small and would benefit from enlargement.  The authors can consider graphing all 3 cell types on one axis, with the appropriate parameters listed in a separate table or on the sides.  The panel designations (A,B,C) are entirely too small.

Made the figures bigger to help see them.

  1. the introductory paragraph on CRISPR should be merged with the previous paragraph about alternative approaches.

Merged the two paragraphs together

  1. Peptoids are only one of numerous types of AMP-mimetic compounds which act through membrane disruption and show therapeutic potential.  These should be at least mentioned as alternatives in the introduction with appropriate references.

We added in other AMP-mimetic compounds with appropriate references to the introduction.

  1. The figure legends are a bit confusing regarding the cell types.  Either the types of cells used should be more prominent in the legends (i.e. capitalized and parenthetical references to the panels) and/or included in the figures themselves.  

The types of cells are mentioned in the figure legends based off the part of the graph. For example, a – L2 cells; b – A549 cells; c – HULEC-5a.

  1. The authors should include a structure of TM9.

Added in a structure into the supplemental (Figure S1)

Reviewer 3 Report

Comments and Suggestions for Authors

1. General comments

In the manuscript, the antiviral activity of a new peptoid TM9 against three coronavirus strains was evaluated, and it was demonstrated that the therapeutic window is large enough to allow use of TM9 for treatment. The result represents an advance in the understanding a strategy of TM9 utilization for combatting coronavirus infections.

2. Major revision

1)  Although three coronavirus strains generated infection in mice after 15 min of exposure by aerosol using the Madison aerosol chamber, further studies are needed to evaluate the therapeutic efficacy of TM9 in vivo against three coronavirus strains.

2) It is recommended to show the structure, measured and calculated (theoretical) molecular mass of TM9.

3)  Line 219: Revise “(EC50) was 6.68 μg/mL, yielding a therapeutic index (CC50/EC90) of 2.62” to “Revise “(EC50) was 4.47 μg/mL, yielding a therapeutic index (CC50/EC90) of 3.92”.

4Fig. 3

a) It is essential to explain Fig. 3d.

b) It is essential to show the data (0.01-5 µg/mL) of cell viability (%) in Fig. 3c.

Author Response

1. General comments

In the manuscript, the antiviral activity of a new peptoid TM9 against three coronavirus strains was evaluated, and it was demonstrated that the therapeutic window is large enough to allow use of TM9 for treatment. The result represents an advance in the understanding a strategy of TM9 utilization for combatting coronavirus infections.

2. Major revision

1)  Although three coronavirus strains generated infection in mice after 15 min of exposure by aerosol using the Madison aerosol chamber, further studies are needed to evaluate the therapeutic efficacy of TM9 in vivo against three coronavirus strains.

We agree. The suggested studies, although outside the scope of the presented work, are in the planning phases, since animals models for coronavirus infections are time consuming and costly.

2) It is recommended to show the structure, measured and calculated (theoretical) molecular mass of TM9.

Added in structure and measured/calculated molecular mass in the supplemental (Figure S1, S2)

3)  Line 219: Revise “(EC50) was 6.68 μg/mL, yielding a therapeutic index (CC50/EC90) of 2.62” to “Revise “(EC50) was 4.47 μg/mL, yielding a therapeutic index (CC50/EC90) of 3.92”.

Fixed the text.

4)Fig. 3

  1. a) It is essential to explain Fig. 3d.

Added in details to explain Figure 3d.

  1. b) It is essential to show the data (0.01-5µg/mL) of cell viability (%) in Fig. 3c.

The data from 0.01 – 5 ug/ml are outliers and were removed as a result.

Round 2

Reviewer 2 Report

Comments and Suggestions for Authors

The authors made the majority of improvements/corrections/enhancements.

Author Response

While I am not 100% positive I know exactly what more was asked for, based off the scoring, I have added more information about the antimicrobial peptide mimics into the introduction with references as appropriate to expand further on various categories.

Reviewer 3 Report

Comments and Suggestions for Authors

1.     Minor revision

It is recommended to add the experimental method concerning Fig. 3D in Materials and Methods.

Author Response

We have added in a line explaining what Figure 3D is in the methods.